# Extraction of Ursolic Acid from Apple Peel with Hydrophobic Deep Eutectic Solvents: Comparison between Response Surface Methodology and Artificial Neural Networks

**DOI:** 10.3390/foods12020310

**Published:** 2023-01-09

**Authors:** Haiyan Li, Yugang Liu, Shiyin Guo, Meng Shi, Si Qin, Chaoxi Zeng

**Affiliations:** Department of Biology and Medicine, College of Food Science and Technology, Hunan Agricultural University, No. 1 Nongda Road, Furong District, Changsha 410128, China

**Keywords:** apple peel, ultrasonic-assisted extraction, response surface methodology, artificial neural network

## Abstract

Extracting ursolic acid (UA) from plant resources using organic solvents is incompatible with food applications. To address this, in this study, 15 edible hydrophobic deep eutectic solvents (HDESs) were prepared to extract UA from apple peel, the extraction conditions were optimized, and the optimization strategies were compared. It was found that the solubility of UA in the HDESs can be 9 times higher than the traditional solvent such as ethanol. The response surface optimization concluded that temperature had the greatest effect on the extraction and the optimized test conditions obtained as follows: temperature of 49 °C, time of 32 min, solid–liquid ratio of 1:16.5 g/mL, respectively. Comparing the response surface methodology (RSM) and artificial neural networks (ANN), it was concluded that ANN has more accurate prediction ability than RSM. Overall, the HDESs are more effective and environmentally friendly than conventional organic solvents to extract UA. The results of this study will facilitate the further exploration of HDES in various food and pharmaceutical applications.

## 1. Introduction

Pentacyclic triterpenes are based on 30 carbon skeletons, consisting of five six-membered rings or four five-membered rings and one five-membered ring, and are formed by the arrangement of epoxide squalene [1], which is one of the classes of triterpenes classified according to carbon skeletons [2]. It can be used as an active ingredient in the treatment of diabetes and its complications [3], as antibacterial agents, antibiofilm agents [4], is considered a potential drug for cancer, viruses, bacteria or protozoan infections [1]. Pentacyclic triterpenoids such as oleanolic acid, glycyrrhizic acid, glycyrrhetinic acid, ursolic acid (UA), betulin, and lupeol show a variety of biological activities [3].

As a kind of lipophilic natural pentacyclic triterpenoids, UA has anti-inflammatory [5], antibacterial, antioxidant [6], anti-fibrosis [7], and antiviral activities [8]. UA can induce autophagy to treat acute kidney injury [9], reduce cholesterol activity, and play a beneficial role in the regulation of intestinal microorganisms [10]. UA is present in many herbs, leaves, flowers, vegetables [11], and fruits such as lingonberries, cranberries, olives, loquat fruit, apples, etc. As far as we know, apple juice produces much unnecessary waste (apple pomace) during the production process [12]. Apple peels, as a waste by-product, contain phytochemicals such as UA [13]. For environmental considerations, it is, therefore, possible to utilize this unwanted waste by extracting UA from apple peels.

Traditional extraction methods of triterpenoids include impregnation, Soxhlet extraction, and hot reflux extraction. The traditional solvents for extracting pentacyclic triterpenes are mainly organic solvents, including methanol, ethanol, and hexane [11]. Kikowska et al. extracted pentacyclic triterpenoids from papaya callus with methanol [14]. Lourenço et al. extracted pentacyclic triterpenoids from birch bark with n-hexane and triterpenes in *eucalyptus globulus* with dichloromethane [15]. Xia et al. extracted pentacyclic triterpenoid UA and oleanolic acid from *ligustrum lucidum* fruit with 80% ethanol [16]. However, these traditional solvents have the disadvantages of being volatile, flammable, explosive, and non-environmental protection, so it is necessary to find a green, non-toxic, and low-volatile solvents to replace these traditional solvents.

Deep eutectic solvent (DES) is a new type of solvent which is obtained by mixing hydrogen donor (HBD) and hydrogen acceptor (HBA) in a certain molar ratio [17]. DES has the advantages of environmental friendliness, low volatility, low cost, easy preparation, sustainability, and biodegradability [18,19]. However, most early studies focused on hydrophilic DES, which limits the extraction of hydrophobic substances. In 2013, Abbott et al. first proposed a hydrophobic deep eutectic solvent (HDES) composed of capric acid and quaternary ammonium salts, which broadened the application of DES in extraction [20]. The extraction of bioactive substances with HDESs instead of traditional organic solvents has recently become a new trend. Ma et al. prepared five HDESs and then screened HDES for liquid-liquid microextraction of diphenylamine in fruits using n-Caprylic acid as HBD and menthol (Men) as HBA [21]. Khare et al. prepared 28 HDESs for achieving green extraction of mushroom ergosterol and obtained maximum extraction efficiency using Men: pyruvic acid [22]. Zhu et al. prepared a variety of HDESs using liquid-liquid microextraction combined with HPLC to determine eight synthetic pigments in beverage samples [23]. Silva et al. used HDES based on Men and thymol (Thy) instead of traditional organic solvents to extract pentacyclic triterpene from eucalyptus globulus bark. They found higher extraction rates than those extracted with methylene chloride [24]. However, the process optimization of this green extraction method is less systematically evaluated.

Response surface methodology (RSM) is an optimization method that uses mathematical and statistical methods to construct an optimal condition for assessing the independent variables’ significance and obtaining predicted response values [25]. Artificial neural networks (ANN) is a neural network inspired by the human brain capable of dealing with nonlinear problems without exact relationships, creating a nonlinear modeling approach between the independent and dependent variables, a highly interconnected structure [26,27]. There is accumulating evidence indicating that, compared with RSM, ANN can obtain more superior prediction results and optimization capabilities through its advantages, and the predicted results are closer to the actual values of the test [28,29]. Therefore, it is also necessary to systematically compare these two optimization methods in extracting bioactive substances with HDESs as a new generation of green solvents.

This experiment aims to extract UA from apple peel using natural Men-based HDESs and systematically compare the test effects of the two optimization methods. Firstly, the type of HDESs with the highest solubility was screened by testing the solubility of UA. Then the molar ratio of HBA and HBD of HDES was further screened to select the HDES with the highest solubility of UA. A single-factor test determined the optimum temperature, time, and solid–liquid ratio of HDES, and then the response surface optimization test was used to obtain the optimum test conditions. This study also used ANN to calculate several sets of data from response surface tests and compares the predictive power of these two methods by comparing their adjust coefficients of determination (R_adj_^2^) and root mean square errors (RMSE).

## 2. Materials and Methods

### 2.1. Materials

The apples (Red Fuji, Yantai, China) were purchased from a local supermarket. UA standard and decanoic acid (purity 99%) purchased from Shanghai Aladdin Biotechnology Company. Menthol (purity 95%) and thymol (purity 99%) were purchased from Wuhan Lanabai Pharmaceutical and Chemical Company. Methanol was purchased from China Ferton Reagent Company (HPLC grade). Pyruvic acid (purity 98%), levulinic acid (purity 99%) and propionic acid (purity 99.5%) were purchased from Shanghai Macklin Biotechnology Co., Ltd. (Shanghai, China). Formic acid (purity 98%), acetic acid (analytical purity), lactic acid (analytical purity), n-butanol (purity 99.5%), and anhydrous ethanol were purchased from China Pharmaceutical Group Chemical Reagent Co., Ltd. (Shanghai, China). Isopropanol (purity 99.7%) was purchased from Tianjin Institute of Chemical Reagents. Phosphoric acid (purity 90%) was purchased from Tianjin Comiou Chemical Reagent Co., Ltd. (Tianjin, China).

### 2.2. Preparation of HDESs

Based on the method of Peng et al., the HBD and HBA were placed in a reagent bottle and stirred at 80 °C for 30 min at 1 g with a collector thermostatic magnetic stirring bath, then a homogeneous liquid was formed and placed in at dry room temperature and set aside [30]. Silva et al. proposed the extraction of triterpenoid acids with Men and Thy [24]. Cai et al. proposed that DESs based on choline chloride and various hydrogen bond donors were chosen to extract UA, and they found that using organic acids as HBD could improve the extraction efficiency of UA [31]. The types of HBA and HBD in this study are shown in Table 1.

### 2.3. Viscosity of HDESs

Referring to Alhadid et al., the viscosity of all HDESs was measured using a Kinexus advanced rotational rheometer (Kinexus pro+, Malvern, UK) at room temperature 30 °C and atmospheric pressure with a gap of 0.1 mm and a shear stress of 1 Pa [36]. The test stopped after the shear viscosity was stabilized, and the data were averaged from the three data after stabilization. The viscosity of HDES with the best solubility at 30 °C, 40 °C, 50 °C, 60 °C, and 70 °C was then tested.

### 2.4. Solubility of UA

#### 2.4.1. Screening Different Types of HDESs

The solubility of UA in 17 HDESs was determined according to the method proposed by Silva et al., with slight modifications [24]. The excess UA and HDES were swirled in a constant temperature magnetic stirrer at a speed of 1 g and a temperature of 60 °C for 30 min (at least three independent samples were prepared for the determination of solubility values and standard deviations). After the supersaturated undissolved UA and the supernatant were separated, 50 μL supernatant sample was taken. The sample was diluted with 495 μL methanol and permeated 0.22 μm through the membrane. The Agilent 1100 liquid chromatography system was used for HPLC determination. The system has a four-component solvent delivery system, an automatic sampler and a diode array detector (DAD). The detection wavelength is 210 nm. The separation is performed on a YMC-Pack ODS-A (250 × 4.60 mm) column with a column box and an automatic sampler maintained at 30 °C. The mobile phase was chromatographic grade methanol, and 0.1% phosphoric acid (92.5%:7.5%, *v*/*v*), and the flow rate was 1 mL/min. UA standard solutions of 20, 40, 80, 160, 320, and 640 μg/mL were prepared, and then, the standard curve was plotted with UA concentration as the horizontal coordinate and peak area (Y, mv/s) as the vertical coordinate (X, μg/mL). Then, the regression equation was established (Equation (1)), and the coefficient of determination R^2^ is 0.9997.
(1)Y=0.8248X−33.913

The solubility was calculated by the following equation:(2)$$\mathrm{Solubility}\;\left(\mathrm{mg}∕\mathrm{mL}\right)=\frac{C_{\mathrm{sol}}\times D}{1000}$$
where C_sol_ represents the concentration of UA of the tested HDES solubility, D is the dilution multiple of the sample, and 1000 represents the unit conversion.

The HDES with the highest UA solubility was selected by solubility test, and then, the ratio of HBA and HBD of the best HDES were screened.

#### 2.4.2. Screening the Ration of HBA and HBD of HDESs

The HDES with the highest UA solubility among the tested HDESs were further selected to test the UA solubility in HDES under different HBA and HBD molar ratios. The molar ratios of the selected HDESs are shown in Table 2. Except for the HDES of MT 4:1, which formed a clear colorless solid at room temperature, all other HDESs formed a uniform and stable clear colorless liquid, indicating that different ratios of HDESs were successfully prepared.

### 2.5. HDES Extracts UA from Apple Peels

The apple peels were placed in an oven and baked at 60 °C for 24 h to constant weight. The dried product was crushed and passed through a 60 mesh sieve. After passing through the sieve, the apple powder was baked at 60 °C for 1 h to prevent water reabsorption during the sieving. The 1 g sample was accurately weighed and added into 10 mL HDES solution and extracted with ultrasonic cleaner (KS3200DE, Kunshan Jielimei Ultrasonic Instrument Co., Ltd.) (Kunshan, China) at a power of 100 W and temperature of 50 °C for 30 min. The extract was placed in a high-speed centrifuge at 5550× *g* for 10 min, take 0.2 mL of supernatant with 1.8 mL of methanol for dilution, passed through a 0.22 μm membrane and then measured by HPLC to calculate the extraction yield (EY) of UA with the following formula.
(3)$$\mathrm{EY}\;=\frac{Mass\;of\;ursolic\;acid}{Mass\;of\;apple\;peel}\times 100\%$$

### 2.6. Single-Factor Test for UA Extraction by HDES

According to the screening of Section 2.5, HDES with the highest solubility were selected to extract UA from apple peels. The optimal extraction time was obtained by changing the ultrasonic extraction time (15, 30, 45, 60, and 75 min) at the controlled temperature of 50 °C and the solid–liquid ratio of 1:14 g/mL. Then, the effect of extraction temperature (30, 40, 50, 60, and 70 °C) on extraction efficiency was explored when the solid–liquid ratio was 1:14 g/mL, and the extraction time was 30 min. Finally, the influence of different solid–liquid ratios (1:14, 1:16, 1:18, and 1:20 g/mL) on the extraction efficiency was explored at the time of 30 min and temperature of 50 °C. The most appropriate single factor conditions were determined, which provided conditions for response surface optimization experiment.

### 2.7. Response Surface Optimization of Extraction Process

Based on the results obtained from the single-factor test in Section 2.6., the optimal extraction conditions were selected, and a three-factor, three-level Box–Behnken design (BBD) response surface test was designed. The response surface optimization data were processed with design expert 13 to optimize the extraction conditions in Section 2.6. further, and the response surface optimization table (Table 3) was shown as follows:

### 2.8. ANN Modeling of Extraction Process

ANN comprises four parts: input, hidden layer, output layer, and output. The input layer of this experiment was composed of time, temperature, and solid–liquid ratio, and the output layer was the extraction yield of UA. The structure of ANN is shown in Figure 1.

In order to obtain better-predicted values and prevent model overfitting, the data is normalized, and the normalized formula is as follows.
(4)$${\;x}_{\mathrm{normalized}\;\mathrm{value}}=\frac{y_{\mathrm{actual}\;\mathrm{value}}-{\;y}_{\mathrm{minimum}\;\mathrm{value}}}{y_{\mathrm{maximum}\;\mathrm{value}}-y_{\mathrm{minimum}\;\mathrm{value}}}$$

All the data involved in the training of ANN are derived from 17 sets of data from the RSM optimization experiment, in which the training set accounts for 70%, the validation set, and the test set account for 15%, respectively. The number of hidden layer neurons is calculated as follows:(5)$$h=\sqrt{n+l}+a$$
where h is the number of neurons in the hidden layer, n is the number of network nodes in the input layer, l is the number of network nodes in the output layer, and a is a constant of 0~10.

### 2.9. Comparison of Predictive Capability of RSM and ANN

The accuracy of performance was evaluated by comparing R_adj_^2^ and RMSE between RSM and ANN. The calculation formula of R_adj_^2^ is as follows:(6)$$R_{\mathrm{adj}}{}^{2}=1-\frac{\sum_{1}^{n}{{(Y}_{\mathrm{cal}}-Y_{\mathrm{Exp}})}^{2}}{\sum_{1}^{n}{{(Y}_{\mathrm{Avg}}-Y_{\mathrm{Exp}})}^{2}}$$

The calculation formula of RMSE is as follows:(7)$$\mathrm{RMSE}=\sqrt{\frac{\sum_{1}^{n}{{(Y}_{\mathrm{cal}}{\;-\;Y}_{\mathrm{Exp}})}^{2}}{n}}$$
where Y_cal_ is the calculated value of RSM or ANN, Y_Exp_ is the true value of the experiment, Y_Avg_ is the average value of the experiment, and n is the number of experiments for 17.

### 2.10. Statistical Analysis

All experiments were conducted in triplicate. RSM design of experiments and model fit was performed in Design Expert 13 (Stat-ease, Inc., Minneapolis, MN, USA), ANN modeling was performed in MATLAB (R2018a, The MathWorks, Inc., Massachusetts, MA, USA), and the other statistical analysis was performed using GraphPad Prism 9 (GraphPad Software, San Diego, CA, USA). One-way analysis of variance (ANOVA) with Tukey’s test was used to determine the significant differences (*p* < 0.05) between the means was performed using IBM SPSS statistic 26.0 (International Business Machines Corporation, Armonk, NY, USA) software. All values are expressed as means ± standard deviations (S.D) unless stated otherwise.

## 3. Results and Discussion

### 3.1. Physical Properties of HDESs

This study determined the viscosity of 16 types of HDESs and the viscosity of the HDES with the most appropriate solubility at 30–70 °C to provide a reference for the solubility screening process. For most of the DES, their viscosity at room temperature is generally above 0.1 Pa [37]. High viscosity will affect the mass transfer during extraction and reduce the extraction efficiency, so the viscosity of HDES needs to be one of the reference factors when screening solvents. It can be seen from Figure 1a that increasing the chain length of fatty acids will increase the viscosity. From Men: formic acid, Men: acetic acid, Men: propionic acid, Men: capric acid, the viscosity of these fatty acids increases with the increase of chain length. This phenomenon also appeared in Men: isopropanol and Men: n-butanol. With the increase in carbon chain length, the viscosity also increased. The viscosity of binary eutectic is determined by hydrogen bond, van der Waals force, and electrostatic interaction [37]. Men: lactic acid has the highest viscosity, which may be enhanced by force formed between Men and lactic acid. Men: isopropanol has the lowest viscosity, probably isopropanol, as an organic alcohol, has a low viscosity of its own. Secondly, HDES composed of isopropanol and Men may have a low viscosity due to the low interaction force. The other viscosity of HDES are Men: levulinic acid > Men: pyruvic acid.

On the one hand, the physical properties of HDES can be changed by changing the size of the molar ratio between HBA and HBD, where the viscosity has different variations [38]. On the other hand, the viscosity of HDES is influenced by the size of cations, stacking and intermolecular interactions [39]. As can be seen from Figure 2a, HDES composed of Men and Thy, with this process of molar ratio from 2:1 to 1:6, the viscosity of HDES gradually increased with the increase of the molar proportion of HBD. However, at 1:8 molar ratio the viscosity of HDES shows an increasing trend.

The viscosity of DES decreases with increasing temperature [40]. As shown in Figure 2b, the viscosity became lower while increasing the temperature of HDES with MT 1:6 from 30 °C to 70 °C.

In summary, although the viscosity of HDES prepared in this experiment is higher than that of traditional organic solvents, the viscosity of hydrophobic DES is lower than DES. In general, the extraction process is carried out at temperatures above room temperature, so the viscosity of HDES decreases with temperature increase. Therefore, to selecting such HDES as an alternative to traditional organic reagents is feasible.

### 3.2. Solubility of UA

In this study, 16 kinds of Men-based HDESs were prepared to determine the solubility of UA, where the composition of HBA and HBD and the molar ratio of HBA and HBD are shown in Table 1 and Table 2. In order to select HDES with the highest UA solubility, solubility tests were performed on the 16 solvents, and all solubility tests were done at 60 °C. As can be seen from Figure 3a, MT 1:1 has the highest solubility among all HDESs, probably because Men and Thy have more similar polarity to UA, making UA better soluble in this HDES.

Among these solvents from HDES 1 to HDES 4, except Men: formic acid, the solubility of UA decreases as the alkyl chain increases. It is probably because as the alkyl chain increases, the steric effects of HDES composed of alkyl and Men increases. Moreover, it is known from Section 3.1. that the viscosity of HDES increases as the alkyl chain increases, and the high viscosity affects the mass transfer, thereby reducing the solubility of UA. The solubility of HDES 5 and HDES 6 also shows a tendency to decrease with the increase of carbon chains. The effect of branched chains on solubility is negligible when comparing the length of carbon chains. HDES 7, which is composed of Men: pyruvate, has the lowest solubility, probably because pyruvate contains a carbonyl group, which leads to its lower solubility. Men: lactate and Men: levulinic acid also showed higher solubility, but MT 1:1 had higher solubility than all other HDESs.

After preliminary solubility screening, it was determined that MT1:1 had the highest solubility, and then the molar ratio of HBA and HBD was further screened. As can be seen from Figure 3b, UA solubility increases with the increase of the molar ratio of HBD, reaching a peak at MT 1:6. Then, a decreasing trend started at MT 1:8. This is closely related to the viscosity of HDESs; as the proportion of HBD increases, the viscosity of HDES decreases. The low viscosity can accelerate the mass transfer between UA and HDES, thus increasing the solubility of UA.

In general, the solubility of the above HDESs prepared was more than nine times higher than that of traditional solvents such as ethanol. Compared with the traditional organic reagent, the solubility of HDESs is more considerable. Therefore, HDES with the highest solubility of MT 1:6 was selected for the following extraction experiments.

### 3.3. Single Factor Experiment

According to the results in Section 3.2, HDES of MT1:6 was used as the solvent for extracting UA from apple peels. Figure 4a shows the effect of time on the extraction yield. The results showed that with the increase of time, the extraction yield of UA increased gradually and reached the highest extraction yield in 30 min, then decreased with the further increase of extraction time. The change in UA extraction yield may be that increasing the time can increase the leaching of the extracted extract within a certain period. However, with the increase of ultrasonic time, the extraction temperature will rise, resulting in the destruction of UA after a long exposure to ultrasonic irradiation, causing the yields to decrease [41].

Figure 4b shows that the extraction yield reached the highest UA extraction yield at 50 °C, and then with the increase in temperature, the extraction yield showed a downward trend. On the one hand, viscosity decreases with increasing temperature. As mentioned in Section 3.2, increasing the temperature can reduce the viscosity of MT 1:6 HDES, and low viscosity can make UA more fully in contact with HDES, making the yield higher. On the other hand, at a temperature of 60 °C or 70 °C, as mentioned above, prolonged exposure to ultrasonic irradiation and high temperature will lead to the destruction of UA.

Figure 4c demonstrates the effect of the solid–liquid ratio on the extraction yield of UA. Within a specific range, the increase of extraction solvent can improve the extraction yield of UA. The UA extraction yield reached the highest value at 1:16 g/mL but decreased with the increase of the solid–liquid ratio to 1:20 g/mL. It may be because too much solvent will play a dilution role. In the process of ultrasonic-assisted extraction, the too high solid–liquid ratio will reduce the cavitation of ultrasonic [42]. Moreover, too much solvent will dissolve more impurities, compete with UA for dissolution, and reduce the extraction yield.

The most appropriate extraction conditions were initially screened for subsequent response surface optimization based on single-factor time, temperature, and solid–liquid ratio. The most appropriate conditions were time 30 min, temperature 50 °C, and the solid–liquid ratio of 1:16 g/mL.

### 3.4. Optimization Method

#### 3.4.1. Response Surface Methodology

Response surface optimization is an excellent method to reduce the number of experiments to optimize the test. In this experiment, the most appropriate experimental conditions were determined by single factor test: time 30 min, temperature 50 °C, and solid–liquid ratio 1:16 g/mL. Then, according to the results of the single-factor experiment, the response surface optimization experiment of three factors and three levels of BBD was designed, and 17 test combinations were obtained, as shown in Table 4.

The correlation between the experimental data and the model was analyzed based on the ANOVA. The ANOVA table from Table 5 shows that the model *p* < 0.05, so it can indicate that the model is significant, while *p* > 0.05 for the lack of fit indicates that it is not significant. In general, C.V. is used to measure the reproducibility of model [28]. The C.V. of this model is 3.80% less than 10%, which indicates that the model is well reproduced. We can also see that the solid–liquid ratio has the most significant effect on the extraction yield of UA (*p* < 0.05). In contrast, temperature and time have no significant effect on the extraction yield of UA (*p* > 0.05). According to the size of the F-value, it can be seen that the solid–liquid ratio has the most significant effect on the extraction yield, then temperature, and finally, time. The R^2^ in this experiment was 0.9536, and the adjusted R^2^ was 0.8940 (>0.8), indicating that the experimental data fit well with the second-order polynomial equation. The multiple regression analysis of the data resulted in a second-order polynomial fit regression equation for UA extraction yield, which can be used to calculate UA extraction yield, and the equation is as follows:
(8)$$\begin{array}{l} Y=-11.44037+0.041617A+0.160512B+1.01406C-0.000162\mathrm{AB}+0.000492\mathrm{AC}+\\ 0.00075\mathrm{BC}-0.000647A^{2}-0.001703B^{2}-0.032312C^{2} \end{array}$$
where Y is calculated value of response surface optimization. A, B, and C are time (min), temperature (°C), and the solid–liquid ratio (g/mL), respectively.

According to the results of ANOVA for the Quadratic model, the F and *p*-values of AB (time and temperature) were 0.9171 and 0.3701, respectively. The F and *p*-values of AC (time and solid–liquid ratio) were 0.3393 and 0.5785, respectively. BC’s F and *p*-values (temperature and solid–liquid ratio) were 0.3509 and 0.5722, respectively. AB has the most considerable F-value and the smallest *p*-value. In addition, it can be seen from Figure 5 that the surface of Figure 5a is steeper, indicating that the interaction of AB is greater. BC and AC have small *p*-values and large F-values. According to Figure 5b,c, it can be seen that their surfaces are not very steep compared to AB, indicating that the interaction is not apparent.

According to the response surface optimization, the predicted reaction conditions were obtained: time 32.334 min, temperature 49.238 °C, and solid–liquid ratio 1:16.514 g/mL. The calculated UA extraction yield was 1.544%. The optimal reaction conditions were calculated as follows: time 32 min, temperature 49 °C, and solid–liquid ratio 1:16.5 g/mL. The yield obtained by the final verification was 1.566%. Siani et al. extracted UA from dry apple peel with aqueous and hydroethanolic to a content of about 6.12 μg/g [43], which is lower than the content of UA in this study.

#### 3.4.2. ANN Modeling

ANN is a multilayer feedback network model with a feed-forward back propagation algorithm. The ANN model in this study used the Levenberg–Marquardt algorithm in training the data. A two-layer feed-forward network with hidden sigmoid neurons and linear output neurons (fitnet), fits multidimensional mapping problems in extraction prediction arbitrarily well.

In this study, a trial-and-error approach was used to train the ANN model one after another, stopping the ANN training when the mean square error (MSE) value is the minimum and R is the maximum. The statement that it is not limited when choosing the number of neurons in the hidden layer is not true [44]. The number of neurons is not necessarily the more significant. The excessive number of neurons will lead to overfitting the model. According to this principle, an optimization of the ANN model was performed to determine the number of neurons of the ANN, and the final number of neurons obtained is the best when it is 6. Therefore, the structure of the ANN model for this experiment is 3–6–1. That is, the input layer is 3, the output layer is 2, and the number of neurons is 6.

When the neuron node was determined to be 6, RMSE and R were calculated according to Formulas (6) and (7). The MSE of the training set is 2.22 × 10^−6^, the MSE of the validation set is 0.000558, and the MSE of the test set is 0.00256. The accuracy of the experimental model can be demonstrated by the fact that the training and test set errors are close [26]. As seen in Figure 6, the MSE of the training set, the validation set, and the test set tends to be stable after four iterations, and the MSE difference between the training set and the validation set is slight. Therefore, it is considered that the model can be used to calculate the data of this study.

As shown in Figure 7a–d, after optimization according to ANN, the R of the training set is 0.99992, the R of the validation set is 0.99731, the R of the test set is 0.99899, and the R of all sets is 0.99229.

#### 3.4.3. Comparison of Predictive Capability of RSM and ANN

The predicted and experimental values of RSM and ANN are shown in Table 4. It can be seen from these data intuitively that the predicted values and experimental values of ANN are closer than those of RSM. Then the prediction ability of the two methods was compared according to the size of the statistical parameters R_adj_^2^ and RMSE between RSM and ANN. Generally, the higher R_adj_^2^, the lower RMSE, and the more accurate the prediction model is [26]. The R_adj_^2^ and RMSE of RSM were 0.9535 and 0.03254, respectively. The R_adj_^2^ and RMSE of ANN were 0.9797 and 2.72249 × 10^−5^, respectively. It can be seen that the R_adj_^2^ of ANN is higher than RSM, and its RMSE is far less than RSM. Therefore, it can be proved that the ANN model can better calculate the experimental results than the RSM model, which is related to its general ability to approximate nonlinear systems [28]. In conclusion, it can be concluded that the ANN model is more reliable and accurate than RSM in its prediction capability and can be better used to calculate the test results of the extraction process of bioactive substances.

## 4. Conclusions

This study prepared 17 kinds of menthol-based HDESs composed of different HBA and HBD. This natural hydrophobic solvent was used for the first time to extract UA in apple peels to replace the traditional organic solvent. According to the solubility test, the solubility of UA in HDESs is significantly improved compared to that in the traditional organic solvent. The solubility of UA in ethanol is only 4.16 μg/mL, while the solubility of UA in HDES is up to 36.2 μg/mL. The response surface optimization concluded that temperature had the most significant effect on the extraction, and the optimized test conditions obtained were 49 °C, 32 min, and 1:16.5 g/mL, respectively. In this study, two different prediction methods, RSM and ANN, were used to calculate the values of 17 groups of experiments. According to the analysis of the data parameters in the two models, such as R_adj_^2^ and RMSE, ANN has a better prediction ability than RSM. Therefore, this work has the potential to become the foundation of an efficient technique for the extraction of UA for application in food industry. HDES is a promising extraction solvent, and its significance as a solvent is more than that. More studies need to pay attention to the coexistence of DES in the activity protection and efficacy promotion of plant-active ingredients.

## Figures and Tables

**Figure 1 foods-12-00310-f001:**
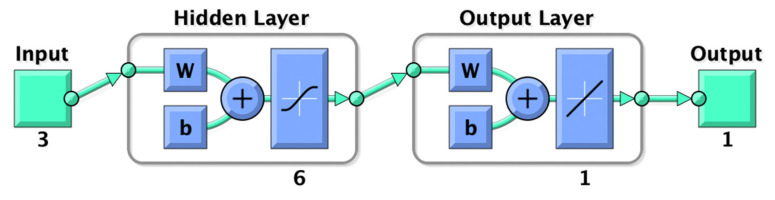
Structure of ANN.

**Figure 2 foods-12-00310-f002:**
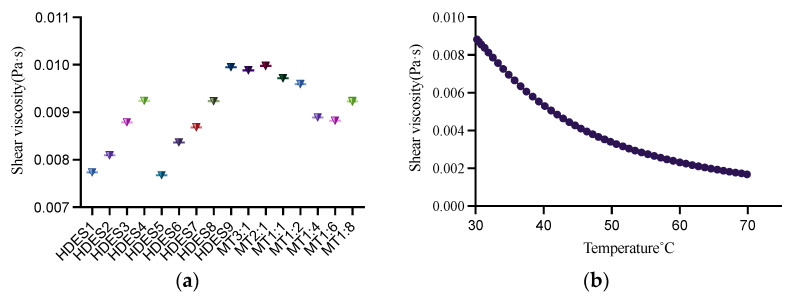
The viscosity of different HDESs at 30 °C (**a**) and the viscosity of MT1:6 at 30~70 °C (**b**).

**Figure 3 foods-12-00310-f003:**
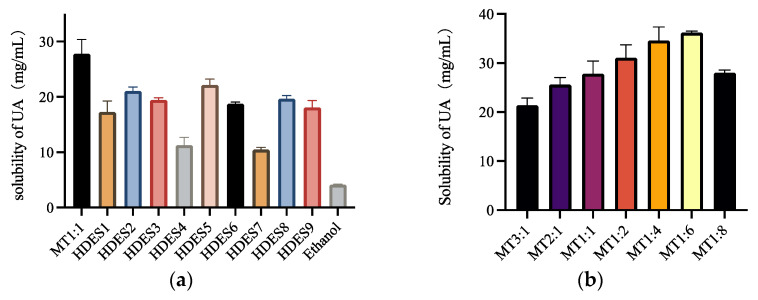
The solubility of UA of HDESs composed of different types of HBA and HBD at 60 °C (**a**), and the solubility of MT1: 6HDES with different HBA and HBD molar ratios at 60 °C (**b**).

**Figure 4 foods-12-00310-f004:**
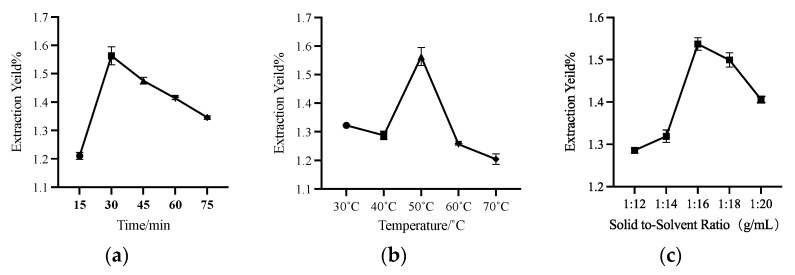
The effects of different time (**a**), temperature (**b**), and solid–liquid ratio (**c**) on the extraction yield of UA.

**Figure 5 foods-12-00310-f005:**
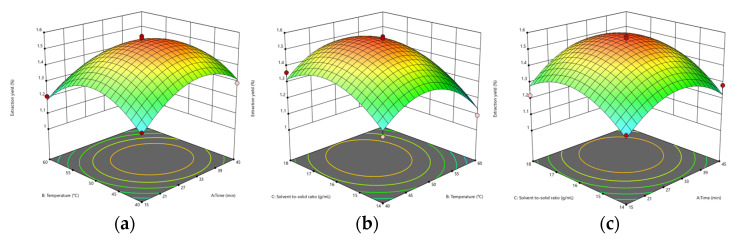
The effect of time and temperature on the yield of UA extraction (**a**), the effect of solid–liquid ratio and temperature on the yield of UA extraction (**b**), and the effect of solid–liquid ratio and time on the yield of UA extraction (**c**).

**Figure 6 foods-12-00310-f006:**
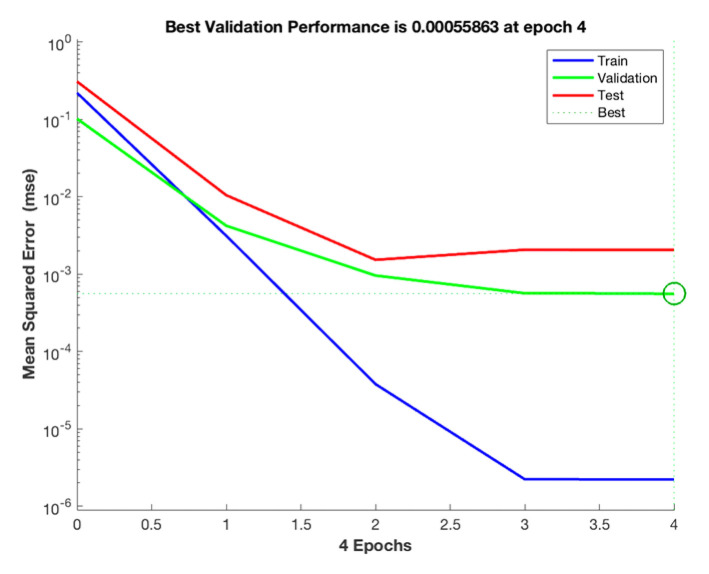
Evolution of MSE during the ANN training phase.

**Figure 7 foods-12-00310-f007:**
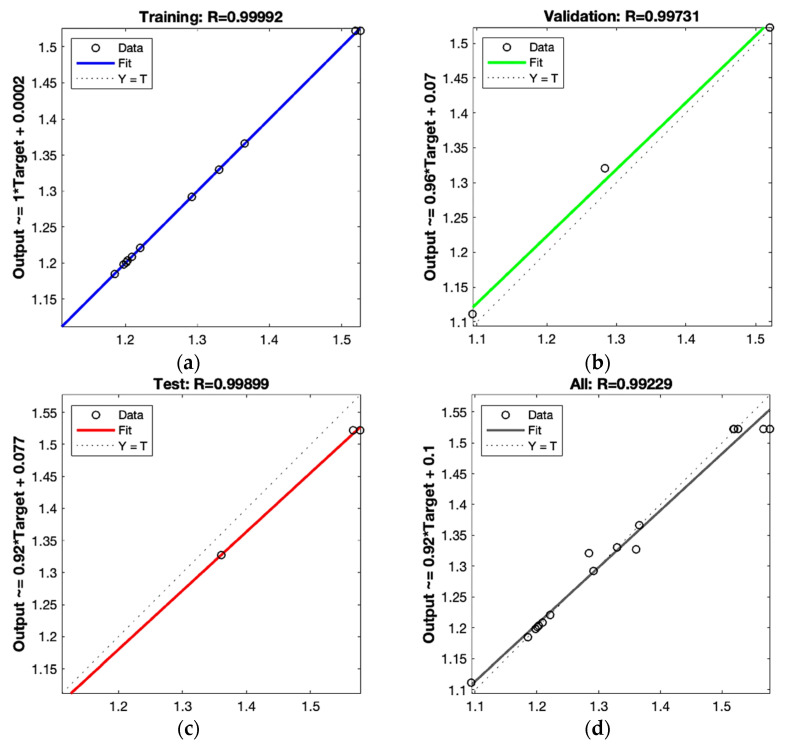
Scatter plot of test data (target) compared with the computed neural network data in the training set (**a**), validation set (**b**), test set (**c**), and all sets (**d**).

**Table 1 foods-12-00310-t001:** The molar ratio and aspect of HDES.

Abbreviation	HBA	HBD	Molar Ratio	Aspect	Reference
MT 1:1	Men	Thy	1:1	Transparent colorless liquid	[24]
HDES 1		Formic acid	1:1	White emulsion	[32]
HDES 2		Acetic acid	1:1	Transparent colorless liquid	[22]
HDES 3		Propionic acid	1:1	Transparent colorless liquid	[32]
HDES 4		Decanoic acid	1:1	Transparent colorless liquid	[33]
HDES 5		Isopropanol	1:2	Transparent colorless liquid	[34]
HDES 6		n-Butanol	1:1	Transparent colorless liquid	[35]
HDES 7		Pyruvic acid	1:2	Transparent yellow liquid	[22]
HDES 8		Lactic acid	1:1	Transparent colorless liquid	[34]
HDES 9		Levulinic acid	1:1	Transparent yellow liquid	[34]

**Table 2 foods-12-00310-t002:** Molar ratio of HBA and HBD.

Abbreviation	HBA	HBD	Molar Ratio	Aspect
MT 4:1			4:1	Transparent colorless solid
MT 3:1	Men	Thy	3:1	Transparent colorless liquid
MT 2:1			2:1	Transparent colorless liquid
MT 1:1			1:1	Transparent colorless liquid
MT 1:2			1:2	Transparent colorless liquid
MT 1:4			1:4	Transparent colorless liquid
MT 1:6			1:6	Transparent colorless liquid
MT 1:8			1:8	Transparent colorless liquid

**Table 3 foods-12-00310-t003:** Factors and levels of response surface analysis.

Levels	Independent Variables		
	A (Time, min)	B (Temperature, °C)	C (Solvent to-Solid Ratio, g/mL)
−1	15	40	14
0	30	50	16
1	45	60	18

**Table 4 foods-12-00310-t004:** Results of the Box–Behnken design (BBD) for the extraction yield of UA and ANN calculated.

Run		Factor		EY/%	RSM Calculated	ANN Calculated
	A	B	C			
1	30	40	18	1.361	1.3168	1.3277
2	30	50	16	1.526	1.5402	1.5225
3	15	60	16	1.209	1.1836	1.2090
4	45	40	16	1.292	1.3136	1.2920
5	30	50	16	1.567	1.5402	1.5225
6	45	60	16	1.201	1.2121	1.2010
7	45	50	18	1.366	1.3833	1.3660
8	30	50	16	1.52	1.5402	1.5225
9	15	50	18	1.221	1.2767	1.2210
10	15	40	16	1.203	1.1880	1.2030
11	30	50	16	1.519	1.5402	1.5225
12	30	60	14	1.094	1.1345	1.1115
13	30	60	18	1.33	1.2938	1.3300
14	45	50	14	1.284	1.2245	1.3209
15	15	50	14	1.198	1.1769	1.1980
16	30	40	14	1.185	1.2173	1.1850
17	30	50	16	1.578	1.5402	1.5225

**Table 5 foods-12-00310-t005:** ANOVA statistics of the quadratic model for the extraction yields of UA.

Source	Sum of Squares	df	Mean Square	F-Value	*p*-Value	
Model	0.3691	9	0.0410	15.99	0.0007	significant
A	0.0122	1	0.0122	4.74	0.0658	
B	0.0054	1	0.0054	2.09	0.1917	
C	0.0334	1	0.0334	13.03	0.0086	
AB	0.0024	1	0..0024	0.9171	0.3701	
BC	0.0009	1	0.0009	0.3393	0.5785	
AC	0.0009	1	0.0009	0.3509	0.5722	
A^2^	0.0891	1	0.0891	34.75	0.0006	
B^2^	0.1220	1	0.1220	47.58	0.0002	
C^2^	0.0703	1	0.0703	27.42	0.0012	
Residual	0.0180	7	0.0026			
Lack of fit	0.0148	3	0.0049	6.17	0.0556	Not significant

## Data Availability

Data is contained within the article.

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
