# Peer review of "Extraction of Ursolic Acid from Apple Peel with Hydrophobic Deep Eutectic Solvents: Comparison between Response Surface Methodology and Artificial Neural Networks"

_foods, 2023, doi:10.3390/foods12020310_

Round 1

Reviewer 1 Report

General comment

The study describes the extraction of ursolic acid (a pentacyclic triterpenoid with potent functional, biological, and pharmacological properties) from food waste (apple peel) with hydrophobic deep eutectic solvents (HDESs). HDESs are more environmentally-friendly and effective than conventional organic solvents. The authors optimized the extraction conditions using a numerical optimization approach based on the Box-Behnken design (BBD) of the response surface methodology (RSM). The extraction process was further modeled using two approaches, RSM and artificial neural networks, and the predictive capabilities of these approaches were compared. The generalized ANN had superior predictive power compared to RSM. Overall, the results showed that HDES had significantly superior solubility of UA (36.2 μg/mL) as compared to conventional extraction solvents, in particular ethanol (4.16 μg/mL), thus demonstrating the applicability of HDES as a possible alternative extraction solvent to conventional organic solvents. The findings herein could propel increased exploration of HDESs for various food and medicinal applications, and consequently stimulate interest in the practice of ‘green chemistry’ amongst bioanalytical scientists.

The research question is relevant and warrants investigation, nonetheless, some technical aspects of the methodology and results are not adequately discussed/described. The authors will need to provide a more adequate description of the RSM optimization approach and ANN model training. Also, the article contains numerous grammatical errors and needs significant English revision. Some specific comments on the manuscript can be found below.

Specific comments

Page 1, line 14: Correct the phrase “Though the response surface optimization…”

Page 1, line 15: The parameters for which the optimized values (i.e., 49ËšC, 32 min, 1:16.5 15 g/mL, respectively) are stated should be named accordingly.

Page 1, line 44: Correct the phrase “Such as, Kikowska et al. extracted pentacyclic…”

Page 2, line 73: I would disagree that ANNs are biological neural networks. ANNs are actually computer algorithms that mimic the biological working of the human brain, which is why they are called “artificial neural networks”.

Page 3, line 107: Provide the exact or approximate stirring speed of the magnetic stirrer used.

Page 5, lines 166-175: Why was Box-Behnken design (BBD) used rather than other RSM approaches such as the central composite design. The authors should provide a clearer description of how the extraction conditions were optimized, and which values are the optimized conditions. On Page 9, lines 304-305: It is stated that “The optimal conditions obtained were time 30 min, temperature 50ËšC, and solid-liquid ratio 1:16 g/mL.” A similar statement is made in Page 10, lines 318-320. How was the optimal extraction condition determined without fitting the optimization model to the experimental data based on the RSM? On Page 11, lines 344-348, two other optimized extraction conditions are presented stating “The prediction of the optimal reaction conditions was finally determined”.

Page 6, lines 103-104: it is stated that “All statistical analysis was performed using Prism 9”, yet the RSM design of experiments and model fit was performed in Design Expert. All statistical packages utilized in the study should be mentioned here including the company's name and country. What software or programme was used for the ANN modelling?

Page 10, line 310: A discussion about the various factor effects should be provided, i.e., the linear, quadratic, and interaction effects of the experimental factors.

Page 10, line 314: Higher resolution images should be provided for figure 5. Also, the equations for each graph should be provided.

The discussion on the RSM results and model fit should be expanded upon. Also, the surface plots should be discussed in more detail.

Page 11, lines 337: The Y variable in equation 8 should be described.

Page 11, lines 349-367: More details on the training of the ANN should be provided, e.g., number of training epochs, batch size, activation function, etc. Were the model hyperparameters tuned? If so, what approach was adopted for tuning of the hyperparameters?

Page 11, lines 360-366: From the results presented herein, it looks like the ANN model might have been overtrained. Generally, when generalizing NNs, it is the case that the networks might be overfitted, as such, a number of approaches are adopted to prevent overfitting such as including a dropout layer or batch normalizations. What measures were taken to prevent overfitting of the model? Also, the authors should provide graphs of the model training loss and validation loss.

Page 12, lines 369-370: Figure 6 contains four separate images. These images should be labeled and a brief description provided in the legends accordingly.

Page 12, line 378: it is stated that the R2 of the ANN is 0.9797, whereas, on Page 11, lines 363-365, it is stated that “after optimization according to ANN, the R of the training set is 0.99992, the R of the validation set is 0.99731, the R of the test set is 0.99899, and the R of all sets is 0.99229.” Which is the correct R2 value for the model. This should be presented clearly. How come about the R2 of 0.9797?

Reviewer 2 Report

- Please rewrite and organize the abstract according to the following context:

A short introduction, hypothesis (aim) of the study, methods, the most important quantitative results, a general conclusion, and future prospective

- The authors should compare the results found with other work/research.

- In the conclusions section, please highlight the future standpoint well.

- Manuscript has grammatical errors, please check.

Reviewer 3 Report

The manuscript presented for evaluation, in accordance with the title, concerns the extraction of ursolic acid from apple pulp using hydrophobic deep eutectic solvents as an alternative to typical solvents used for the extraction of this compound, such as ethanol or acetone. In this context, the topic taken up by the Authors of the manuscript fits into the contemporary research issues.

Ursolic acid (UA) due to its biological activity is a sought-after and valuable compound for the pharmacological industry. It is a compound that is quite common in many plants, but its content is generally low. To the knowledge of the reviewer, due to the very complex molecular structure with 10 chiral centers, ursolic acid has not yet been synthesized in the laboratory. For this reason, some plants and fruit peels have so far been sources of UA. The problem with UA, however, is not its extraction but its purification, because it has a very similar structure to its isomer, and therefore also its physicochemical properties are very similar, which makes its separation difficult.

The reviewed manuscript only mentions that the UA analysis was performed by HPLC. However, neither the conditions of the chromatographic analysis nor the exemplary chromatogram are described. Not to mention the purity of the resulting acid. The analytical part is very limited to one point "The standard curve of UA" wrongly described in the form of unreal units (640g/mL). There is also no comparison of the UA yield obtained by the proposed method with the results obtained using classical solvents. In addition, according to the reviewer, the use of RP-HPLC chromatography and not precipitation chromatography in the studies of UA solubility in the obtained solvents is highly problematic. Therefore, the title of the manuscript "extraction of ursolic acid from apple peel" is strongly exaggerated.

Round 2

Reviewer 2 Report

The MS has been significantly improved and I appreciate the authors' efforts to respond to reviewer comments. So, the MS can be published in its current form.

Author Response

Thanks.

Reviewer 3 Report

The manuscript has been sufficiently improved to be published in Foods.

Author Response

Thanks.